# No Wrong Turns: The Simple Geometry Of Neural Networks Optimization Paths

## Abstract

Understanding the optimization dynamics of neural networks is necessary for closing the gap between theory and practice. Stochastic first-order optimization algorithms are known to efficiently locate favorable minima in deep neural networks. This efficiency, however, contrasts with the non-convex and seemingly complex structure of neural loss landscapes. In this study, we delve into the fundamental geometric properties of sampled gradients along optimization paths. We focus on two key quantities, which appear in the restricted secant inequality and error bound. Both hold high significance for first-order optimization. Our analysis reveals that these quantities exhibit predictable, consistent behavior throughout training, despite the stochasticity induced by sampling minibatches. Our findings suggest that not only do optimization trajectories never encounter significant obstacles, but they also maintain stable dynamics during the majority of training. These observed properties are sufficiently expressive to theoretically guarantee linear convergence and prescribe learning rate schedules mirroring empirical practices. We conduct our experiments on image classification, semantic segmentation and language modeling across different batch sizes, network architectures, datasets, optimizers, and initialization seeds. We discuss the impact of each factor. Our work provides novel insights into the properties of neural network loss functions, and opens the door to theoretical frameworks more relevant to prevalent practice.

## 1 Introduction

Despite the theoretical complexity of their loss landscapes, deep neural networks have demonstrated remarkable empirical reliability across a broad range of applications. Blum & Rivest (1992) proved decades ago that neural network training is NP-hard. The intricacy of their loss functions, especially the non-convexity implying potential bad local minima and saddle points, has led to an enduring conundrum concerning the empirical efficiency of stochastic first-order optimization methods for training neural networks.

Numerous studies have strived to reconcile this apparent contradiction, focusing on the behaviors of stochastic gradient descent (SGD) and its variants at local minima and saddle points (Panageas et al., 2019; Jin et al., 2019). The central hypothesis in these works posits that the efficiency of training arises from the ability of these algorithms to navigate complex loss landscapes adeptly and manage non-convexity.

Conversely, other investigations have empirically found loss landscapes to be simpler than their theoretical complexity might suggest (Lucas et al., 2021). Notably, Goodfellow et al. (2015) observed that "in fact, on a straight path from initialization to solution, a variety of state of the art neural networks never encounter any significant obstacles."

Notwithstanding, our current understanding of how neural loss landscapes are empirically simpler than expected remains quite limited. There is yet to emerge a robust mathematical characterization of this empirical simplicity. Consequently, we contend that the theoretical assumptions currently in use fail to accurately capture the objective functions typical in deep learning. This discrepancy is a significant barrier to applying theoretical insights effectively in the optimization of neural networks.

One such common assumption, smoothness, is illustrative of this gap. Despite its popularity, smoothness is encumbered by several limitations: it is computationally intensive to approximate for large

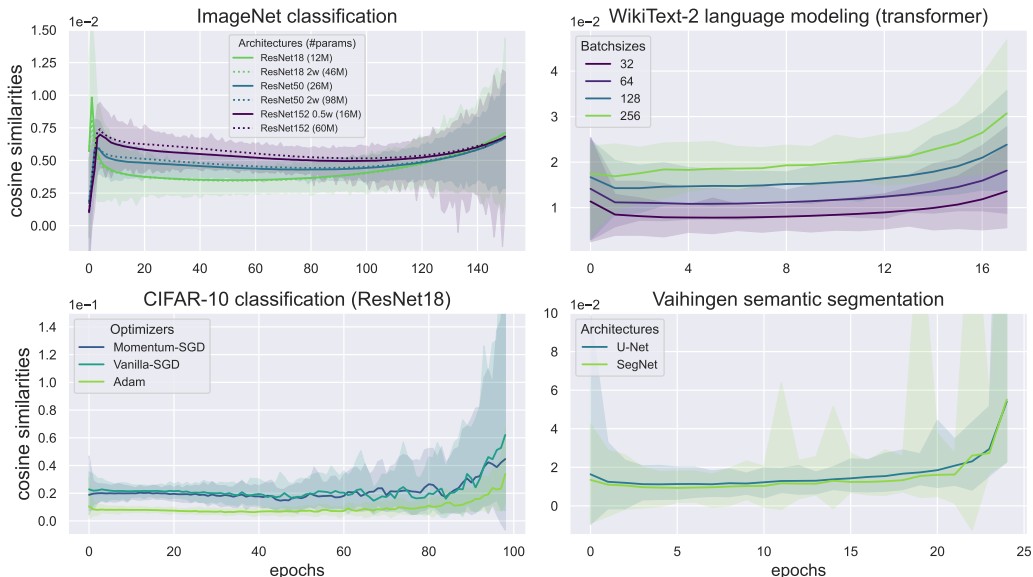

Figure 1: Cosine similarities between the gradients $G_t$ sampled at step $t$ and the difference $(w_t - w_T)$ between current weights and final weights, averaged over each epoch. The shaded regions denote the range from minimum to maximum values observed at each epoch. The results are presented for a selection of scenarios: (top left) varying depths and widths of ResNet on ImageNet, (top right) different batch sizes on WikiText-2 using a Transformer, (bottom left) a range of optimizers on CIFAR-10 using ResNet-18, and (bottom right) distinct architectures on Vaihingen semantic segmentation. This figure highlights the stability of the cosine similarity throughout most of training, suggesting it as a fundamental characteristic of neural loss landscapes.

neural networks, and necessitates additional assumptions such as bounded gradients for theoretical guarantees in stochastic settings (Qian et al., 2019; Shamir & Zhang, 2013) although recent works have tried to discard them (Nguyen et al., 2018; Loizou et al., 2021). Finally, recent findings suggesting certain directional sharpness in neural networks Dinh et al. (2017) call into question the suitability of smoothness as a measure of their simplicity.

To address these issues, our study undertakes an empirical analysis of the geometric properties of the loss function in regions traversed by first-order optimization algorithms. Our focus is on a variant of the quantities involved in the Restricted Secant Inequality (RSI) (Zhang & Yin, 2013) and Error Bound (EB) (Luo & Tseng, 1993), which pertain to the relationship between sampled gradients, current iterate, and final iterate of the optimization sequence. Our findings indicate that these quantities and their ratio exhibit stable, predictable patterns throughout training across diverse settings, thereby quantitatively characterizing the simplicity of neural loss landscape geometry. Furthermore, these quantities offer several advantages over smoothness, including efficient estimations post-training, inherent compatibility with stochasticity due to direct measurement on sampled gradients, and a well-behaved empirical nature that still allows the derivation of theoretical results such as linear convergence or the prescription of learning rate schedules.

**Our key contributions** are as follows:

- We devise an experimental procedure for examining the geometry of optimization paths on common architectures. We assume almost-everywhere differentiability, but not smoothness.
- We execute experiments across a range of realistic deep learning settings, identifying consistently verified properties. For instance, the cosine similarity between the negative stochastic gradient and the direction to the final iterate is almost always positive and exhibits remarkable stability across iterations and epochs.
- We demonstrate how our empirical investigations can inform the prescription of learning rate schedules, aligning with established empirical knowledge.
- We provide an extensive discussion on the implications and limitations of our findings.

Collectively, our work quantifies crucial geometric properties of stochastic gradients along deep learning optimization paths, underlining their importance in understanding neural network optimization and enhancing current methodologies.

## 2 RELATED WORK

Our investigation centers on the application of RSI and EB to enhance our comprehension of the geometric principles governing neural network optimization. Consequently, our work intersects with previous research on neural loss landscapes and the utilization of RSI and EB in optimization.

**RSI and EB:** The study of RSI and EB for first-order optimization is not new. RSI (Zhang & Yin, 2013) has been applied in numerous theoretical works (Yi et al., 2019; Schöpfer, 2016; Yuan et al., 2016; Karimi et al., 2016). EB (Luo & Tseng, 1993) has seen less extensive study (Dmitriy Drusvyatskiy, 2018), possibly due to the dominance of smoothness —a condition stronger than EB— in the field. It should not be confused with error bounds on the distance to a set, a term also prevalent in optimization literature (Qian et al., 2023; Zhou & So, 2015). Both RSI and EB, along with other conditions, were analyzed in Guille-Escuret et al. (2021). Furthermore, it was demonstrated in Guille-Escuret et al. (2022) that gradient descent is optimal for the class of functions defined by this pair of conditions.

**Neural Loss Landscape Geometry:** The intricacies of neural loss landscapes have been a focal point of research since the emergence of deep learning. Efforts have ranged from loss landscape visualizations (Li et al., 2018) to investigations of low loss basin connectivity (Garipov et al., 2018) and linear mode connectivity (Frankle et al., 2020). While prior research has noted the seeming simplicity of loss landscape geometry along optimization paths (Lucas et al., 2021; Goodfellow et al., 2015), these observations often involve straightforward phenomena such as monotonic decrease along linear interpolations. Our work takes this approach a step further by studying quantifiable properties with theoretical implications. Additionally, others have examined the geometric properties of neural loss landscapes in the near-infinite width, or Neural Tangent Kernel (NTK), regime (Jacot et al., 2018; Lee et al., 2019). These studies suggest that neural network training can be approximated by linear dynamics or that the loss surface adheres to the Polyak-Łojasiewicz condition (Liu et al., 2022). Unfortunately, this scenario was found to be distinct from empirical settings (Chizat et al., 2019), although recent studies have delved into the evolution of the NTK under more realistic conditions (Fort et al., 2020). We also note the active research direction regarding the influence of BatchNorm on the optimization trajectory Santurkar et al. (2018b); Ioffe & Szegedy (2015a).

## 3 BACKGROUND

The training of a neural network on a dataset comprised of $n$ examples can typically be formulated as the finite-sum optimization problem

$$\min_{w \in \mathbb{R}^d} \mathcal{L}(w) := \frac{1}{n} \sum_{i=1}^{n} l_i(w), \tag{1}$$

where $w$ are the parameters of the neural network, $\mathcal{L}$ is the empirical risk, and $l_i$ corresponds to the loss function for the $i$-th data sample, for $i = 1, \ldots, n$. We denote the empirical risk with respect to any minibatch $\mathcal{B} \subseteq [n]$ of size $m$ as $\mathcal{L}_\mathcal{B} := \frac{1}{m} \sum_{i \in \mathcal{B}} l_i$. Throughout this work, we assume the loss to be differentiable, but we do **not** require it to be smooth.

We now recall the definitions of $\text{RSI}^-$ and $\text{EB}^+$ as provided by Guille-Escuret et al. (2021). Given an objective function $\mathcal{L}$ with a convex set of global minima $\mathcal{X}^\star$, and letting $w_p^\star$ be the orthogonal projection of $w$ into $\mathcal{X}^\star$,

**Definition 3.1** (Lower Restricted Secant Inequality). *Let $\mu > 0$. $\mathcal{L} \in \text{RSI}^-(\mu)$ iff:*

$$\forall w \in \mathbb{R}^d, \nabla \mathcal{L}(w)^T (w - w_p^\star) \geq \mu \left\| w - w_p^\star \right\|_2^2. \tag{2}$$

**Definition 3.2** (Upper Error Bounds). *Let $L > 0$. $\mathcal{L} \in \text{EB}^+(L)$ iff:*

$$\forall w \in \mathbb{R}^d, \|\nabla \mathcal{L}(w)\|_2 \leq L \left\| w - w_p^\star \right\|_2. \tag{3}$$

The classes of functions $\mathrm{RSI}^-$ and $\mathrm{EB}^+$ are thus defined in the literature as those respecting the above bounds over the entire parameter space. However, in this work, our focus lies not merely on their extremal values but on the local quantities bounded by $\mathrm{RSI}^-$ and $\mathrm{EB}^+$.

For simplicity, we refrain from introducing new terminology, and henceforth denote these quantities as $\mathrm{RSI}(G, w, w^\star)$ and $\mathrm{EB}(G, w, w^\star)$, where $G$ is an oracle for the gradient at $w$. We do not mandate $G$ to be the full gradient of $\mathcal{L}$; it could, for instance, correspond to the gradient $\nabla \mathcal{L}_\mathcal{B}$ with respect to a minibatch $\mathcal{B}$. Similarly, $w^\star$ is not assumed to be a minimum of the objective function. Formally, for any gradient oracle $G$, and any $w^\star \in \mathbb{R}^d$, $w \neq w^\star$:

$$\mathrm{RSI}(G, w, w^\star) := \tfrac{G(w)^T(w - w^\star)}{\|w - w^\star\|_2^2} \quad \text{and} \quad \mathrm{EB}(G, w, w^\star) := \tfrac{\|G(w)\|_2}{\|w - w^\star\|_2}.$$

The ratio between RSI and EB imparts a direct geometrical interpretation:

$$\gamma(G, w, w^\star) := \tfrac{\mathrm{RSI}(G, w, w^\star)}{\mathrm{EB}(G, w, w^\star)} = \tfrac{G(w)^T(w - w^\star)}{\|G\|_2 \|w - w^\star\|_2} = cosine(G(w), w - w^\star),$$

where $cosine(w_1, w_2)$ is the cosine of the angle between vectors $w_1$ and $w_2$.

This ratio, $\gamma$, signifies the alignment between the negative sampled gradient and the direction from $w$ to $w^\star$. When $\gamma$ approaches 1, it indicates a negative gradient strongly directed toward $w^\star$. Conversely, a $\gamma$ close to 0 suggests a gradient almost orthogonal to $w - w^\star$. A negative $\gamma$ indicates a negative gradient directed away from $w^\star$. Additionally, $\gamma$ can be interpreted as the inverse of a local variant of the condition number, $\kappa := \tfrac{\sup \mathrm{EB}}{\inf \mathrm{RSI}}$, which is a measure of the complexity of optimizing $\mathcal{L}$ in prior works (Guille-Escuret et al., 2021).

RSI and EB are intrinsically connected to the dynamics of stochastic gradient descent (SGD). Indeed, the distance to $w^\star$ following an SGD step with step size $\eta$ can be precisely articulated using RSI and EB. For all $w \neq w^\star, \mathcal{B}$,

$$\|w - \eta \nabla \mathcal{L}_\mathcal{B}(w) - w^\star\|_2^2 = \|w - w^\star\|_2^2 - 2\eta \nabla \mathcal{L}_\mathcal{B}(w)^T(w - w^\star) + \eta^2 \|\nabla \mathcal{L}_\mathcal{B}(w)\|_2^2$$
$$= \left(1 - 2\eta \, \mathrm{RSI}\left(\nabla \mathcal{L}_\mathcal{B}, w, w^\star\right) + \eta^2 \, \mathrm{EB}^2\left(\nabla \mathcal{L}_\mathcal{B}, w, , w^\star\right)\right) \|w - w^\star\|_2^2. \tag{4}$$

Consequently, with a step size of

$$\eta^\star := \underset{\eta}{\arg\min} \|w - \eta \nabla \mathcal{L}_\mathcal{B}(w) - w^\star\|_2 = \tfrac{\mathrm{RSI}(\nabla \mathcal{L}_\mathcal{B}, w, w^\star)}{\mathrm{EB}^2(\nabla \mathcal{L}_\mathcal{B}, w, w^\star)}, \tag{5}$$

SGD guarantees

$$\|w_{t+1} - w^\star\|_2 = \sqrt{1 - \gamma(\nabla \mathcal{L}_\mathcal{B}, w, w^\star)^2} \, \|w_t - w^\star\|_2. \tag{6}$$

Furthermore, if $\underset{w,\mathcal{B}}{\inf} \mathrm{RSI}(\nabla \mathcal{L}_\mathcal{B}, w, w^\star) \geq \mu$ and $\underset{w,\mathcal{B}}{\sup} \mathrm{EB}(\nabla \mathcal{L}_\mathcal{B}, w, w^\star) \leq L$ hold for some $\mu > 0$, $L > 0$, then equation equation 4 demonstrates that running SGD with a fixed step size of $\eta = \tfrac{\mu}{L^2}$ will converge to $w^\star$ at a guaranteed rate:

$$\|w_t - w^\star\|_2^2 \leq (1 - \tfrac{\mu^2}{L^2})^t \|w_0 - w^\star\|_2^2, \tag{7}$$

This holds irrespective of how the minibatches are sampled. Under these assumptions, this rate is, in fact, worst-case optimal among all continuous first-order algorithms (Guille-Escuret et al., 2022).

**Experimental Measurement of RSI and EB:** One of the most significant challenges in experimentally measuring RSI and EB lies in the selection of $w^\star$. Even in cases where the objective function admits an unique global minimum, finding it in the context of deep neural networks is computationally infeasible Blum & Rivest (1992). To navigate this complication, we initially train a neural network and subsequently choose the final iterate $w_T$ of the optimization sequence. Given successful training, the sequence will converge to the vicinity of a (local) minimum, and measuring RSI and EB with respect to this minimum will provide insightful understanding of the training dynamics.

Notably, under this procedure, $w^\star$ is dependent on the optimization sequence rather than being predetermined. Therefore, interpreting the ensuing results warrants care, see Section 6.

Considering that saving all gradients and iterates observed during training would be prohibitively resource-intensive, we perform two identical training runs. The first run computes $w^\star = w_T$, and the second run computes RSI and EB along the optimization path. A detailed description of our experimental protocol is provided in Algorithm 1 in Appendix A.1, and we share our code at `https://anonymous.4open.science/r/LossLandscapeGeometry-B7BD/`.

## 4 EMPIRICAL GEOMETRY OF LANDSCAPES ALONG OPTIMIZATION PATHS

Figure 2: Depicted are the trends of RSI (top), EB (middle) and $\gamma$ (bottom) across three different scenarios: image classification on CIFAR-10 with a ResNet-18 (left), image classification on ImageNet with a ResNet-50 (middle) and language modeling on WikiText-2 with a transformer model (right).

Figure 1 offers an initial glance at our results, outlining the behavior of $\gamma$ across four datasets, with variations across architecture, batch size, and optimization technique. Figure 2 presents a more streamlined view on three of these datasets, exhibiting not only $\gamma$ but also RSI and EB on a single run to preserve clarity. To avoid precision issues when $w_t$ approaches $w^\star$, the results from the final epoch have been excluded. Our hyperparameters were initially adjusted to optimize validation accuracy, echoing practical conditions. All experiments were coded in PyTorch Paszke et al. (2019) and detailed descriptions of the specific training configurations, along with final test performances, are available in Appendix A to ensure full reproducibility.

**CIFAR-10 (ResNet-18):** Across the entire training run, not a single iteration exhibits a negative $\gamma$. Even though there are slight fluctuations across epochs, $\gamma$ predominantly remains within the $[0.0075, 0.02]$ range and does not exhibit substantial shifts. While the variance of RSI and EB across iterations tends to increase as training progresses, their mean values largely remain stable.

**ImageNet-1K (ResNet-50):** Except for a few iterations at the very early stage, $\gamma$ remains positive throughout all of training. Moreover, the variance across iterations is notably low until the last epochs. Epoch-wise, RSI, EB, and $\gamma$ increase monotonically, with a sharp rise observed towards the end.

**WikiText-2 (Transformer):** Throughout training, $\gamma$ remains strictly positive and always exceeds $0.05$ after the second epoch. The cosine similarity maintains a remarkable stability, exhibiting only minor variations across iterations and epochs. While RSI and EB show very low variance within epochs, they do increase towards the end of the training period.

### 4.1 FUNDAMENTAL PROPERTIES

Upon careful analysis, we find that the optimization trajectories of deep neural networks exhibit the following major characteristic features:

- The cosine similarity, $\gamma$, is almost always positive.
- $\gamma$ demonstrates notable stability across both epochs and iterations, rarely departing from its (low) average value.
- RSI and EB follow predictable trends, contingent upon whether the model adheres to an interpolation or a non-interpolation regime.

**Interpolation vs Non-Interpolation Regime:** The behavior of RSI and EB are directly tied to how well the final iterate $w^\star$ interpolates the training data. For CIFAR-10, where the model reaches

close to $0$ training loss, RSI and EB retain relatively stable mean values up to the last epochs, which is made possible by stochastic gradients decreasing to $0$ as $w_t$ approaches $w^\star$. Conversely, in scenarios where the model fails to interpolate the training data, such as for ImageNet and WikiText-2, stochastic gradients remain significant. In such a scenario, RSI and EB inevitably rise to infinity as $w_t - w^\star$ approaches zero. This phenomenon is particularly obvious with ImageNet due to the learning rate decay, which induces minuscule distances between $w_t$ and $w^\star$ in the later stages of training. Additional experimental results supporting this interpretation are provided in Appendix C.

**Late Training Behavior:** The results obtained towards the end of training should be interpreted with caution. Besides the previously described phenomenon in the non-interpolation regime, the correlation between sampled gradients and $w_t - w^\star$ increases as the sequence nears its termination. Intuitively, $w^\star$ approximates a minima, and the approximation error becomes significant as iterates get sufficiently close. Further discussion on related implications can be found in Section 6.

**Low Value of Cosine Similarity:** The low values of $\gamma$ empirically encountered are to be expected: if $\gamma$ was stable at reasonably high values, then we would find a near-minima in a small number of steps using SGD, which is notoriously not the case for modern problems. Instead, optimization sequences approach their final iterate at a slow but regular pace. While the stability and positivity of $\gamma$ imply a linear convergence rate, its low value indicate a linear rate close to $1$, similarly to a strongly convex and smooth objective being badly conditioned. A plausible cause for $\gamma$ being small is that the useful signal from generalizable features in sampled gradients is dominated by that of spurious and coincidental correlations.

**Significance:** These observations imply that, despite the well-documented non-convexity of the loss landscapes associated with neural networks and the inherent stochasticity introduced by minibatch sampling, the learning process of neural networks remains remarkably consistent. The networks progress steadily towards their destination throughout the training, with each stochastic gradient contributing valuable information to reach the final model state. With very few exceptions, gradients always point toward the right direction, and training trajectories never take a wrong turn when optimizing the loss function. We find these observations to be particularly remarkable on ImageNet. Given the presence of $1000$ semantic classes (exceeding the batch size) and in excess of $5000$ minibatches per epoch, the consistence of the cosine similarity $\gamma$ throughout entire epochs seems surprising. In addition, Section 6 establishes links between empirically adopted learning rate schedules and RSI and EB. Overall, RSI and EB are powerful tools to capture the elusive simplicity of neural loss landscapes, with empirical properties theoretically guaranteeing linear convergence rates. We thus encourage future works to consider RSI and EB to characterize the classes of objectives encountered in deep learning applications.

We further explore the impact of various factors and provide a more comprehensive substantiation of our findings in Section 5. Following this, we discuss the implications and potential limitations of our observations in Section 6. We also discuss plausible causes in Appendix D.

## 5 INFLUENCE OF TRAINING SETTINGS

**Batch Size:** the top right of Figure 1 delineates the cosine similarities corresponding to batch sizes ranging from $32$ to $256$ on the WikiText-2 dataset. As a complementary experiment, Figure 6 in Appendix B portrays the cosine similarities associated with batch sizes from $64$ to $512$ on the CIFAR-10 dataset. The outcomes of both these experiments consistently reveal a positive correlation between batch size and cosine similarity. This outcome is foreseeable: for two minibatches $\mathcal{B}_i$ and $\mathcal{B}_j$, we have

$$\mathrm{RSI}(\nabla\mathcal{L}_{\mathcal{B}_i}+\nabla\mathcal{L}_{\mathcal{B}_j}) = \mathrm{RSI}(\nabla\mathcal{L}_{\mathcal{B}_i})+\mathrm{RSI}(\nabla\mathcal{L}_{\mathcal{B}_j}), \quad \mathrm{EB}(\nabla\mathcal{L}_{\mathcal{B}_i}+\nabla\mathcal{L}_{\mathcal{B}_j}) \leq \mathrm{EB}(\nabla\mathcal{L}_{\mathcal{B}_i})+\mathrm{EB}(\nabla\mathcal{L}_{\mathcal{B}_j}).$$

It should be noted that the selection of batch size not only affects the measurement of RSI and EB, but it also influences the optimization trajectory and the speed of convergence. Therefore, direct numerical comparisons across different batch sizes ought to be interpreted with caution. Nonetheless, our observations suggest that cosine similarities may scale with the square root of the batch size.

**Optimizer:** Figure 1 (bottom left) illustrates the cosine similarity for three distinct optimizers utilized on the CIFAR-10 dataset. Intriguingly, Adam appears to result in lower cosine similarity values, albeit with reduced variance. We hypothesize that Adam, by amplifying the effective step size along directions with lower curvature, traverses further in flat dimensions, thereby leading to a

reduced alignment compared to SGD. This conjecture is substantiated by Figure 12 in Appendix C, demonstrating that the journey undertaken by Adam indeed surpasses that of SGD in terms of distance. Notably, the employment of a momentum value of 0.9 with SGD does not significantly impact the value of $\gamma$, compared to not using momentum. Prior works also suggest that the optimization methods may affect the geometry of visited regions (Cohen et al., 2021).

**Model Depth and Width:** Our attention now turns to the impact of depth and width on the geometric characteristics of the optimization trajectory, as depicted in Figure 1 (top left). In this experiment, we trained ResNets of varying depth — 18, 50, and 152 layers — with both standard and doubled width. A salient observation is that an increase in depth slightly enhances the cosine similarities, while an increase in width appears to have a comparatively trivial impact. These findings could potentially shed light on the prevalent trend in contemporary neural network designs favouring increased depth over width (He et al., 2016).

## 6  KEY TAKEAWAYS AND DISCUSSION

**Geometrically Justified Learning Rate Schedules:** As established in Equation equation 5, we define the locally optimal learning rate (loLR) as the minimizer of $\|w_t - \eta\nabla\mathcal{L}_{\mathcal{B}_t} - w^\star\|_2$, $\eta^\star(w) = \frac{\text{RSI}(w)}{\text{EB}^2(w)}$. It is important to note, however, that $\eta^\star$ may not necessarily be globally optimal. Indeed, certain

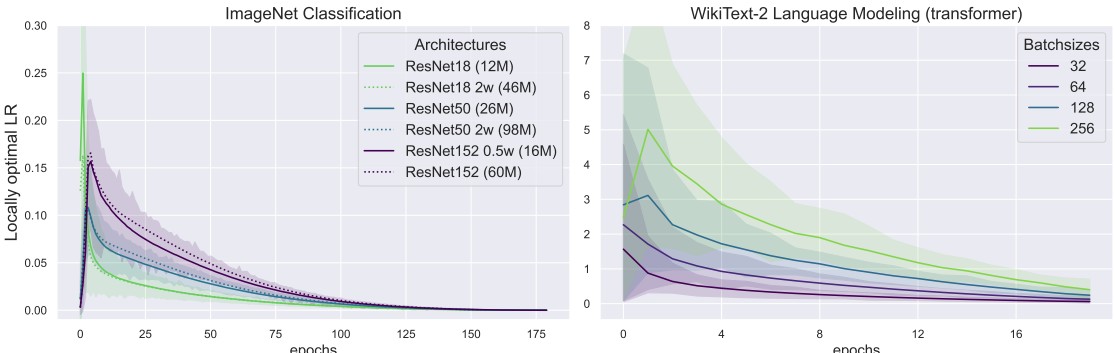

Figure 3: Left panel: The locally optimal learning rate, derived as per Equation equation 5, for various architectures implemented on the CIFAR-10 dataset. Right panel: The locally optimal learning rate, similarly determined, across a spectrum of batch sizes employed on the WikiText-2 dataset.

methodologies may initiate slower but accumulate more information, ultimately leading to faster convergence over a large number of steps. Furthermore, as the measurement of RSI and EB requires the knowledge of $w^\star$, which in turn depends on the learning rate (LR), the expression cannot be utilized to dynamically tune it.

Despite these limitations, we find intriguing parallels between the evolution of the loLR derived from our experiments and the shape of empirically validated LR schedules, as demonstrated in Figure 3. For instance, a widely adopted strategy for training on ImageNet involves a linear warm-up phase of the LR for the initial few epochs, followed by a cosine annealing phase. This pattern is mirrored in our empirical observations on ImageNet, except for a sharper decrease immediately after warmup.

Moreover, the results on WikiText-2 echo two popular practices: linearly decreasing the LR and increasing the batch size over time. These intriguing observations suggest that the geometry of the loss landscape could potentially inform the design of more effective learning rate schedules.

Lastly, the apparent correspondence between loLR and empirical learning rate strategies implies that the efficiency of fixed learning rates may be contingent upon the stationarity of RSI and EB. Similarly, the existence of straightforward and efficient learning rate schedules can be associated with the predictable evolution of these geometrical properties. This strongly reinforces the view that such geometrical attributes play a substantial role in the widespread practical successes of deep learning.

**Biases Induced by Using Final Iterates as Reference Points:** A critical limitation of our experimental approach is the inescapable correlation between $w^\star$ and the optimization sequence. This association must be thoroughly addressed to appropriately interpret our findings.

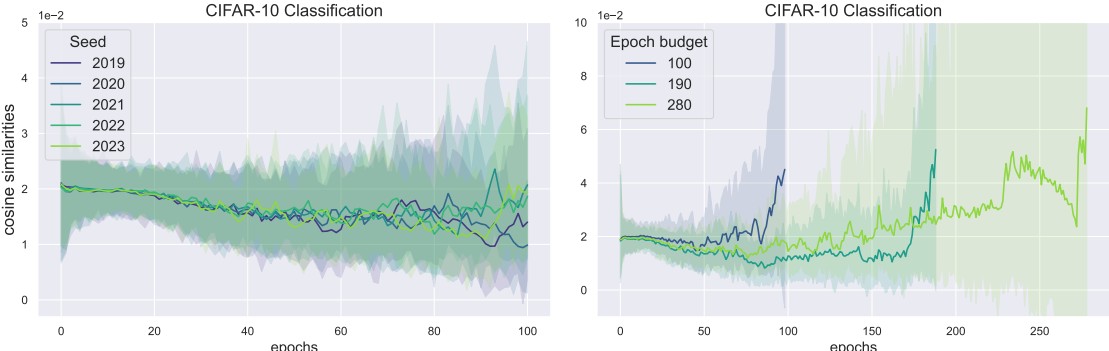

Figure 4: Depiction of cosine similarities during the training of a ResNet-18 on the CIFAR-10 dataset, with variations in (left) initialization seed and (right) epoch budget.

• **Initialization:** Firstly, RSI and EB may represent local properties of the loss landscape, and could be dependent on the initialization region. However, this possibility is refuted by the left panel of Figure 4, which demonstrates minimal variation in $\gamma$ measurements across different random seeds.

• **Epoch Budget:** Secondly, our results might be influenced by the particular moment when we terminate the optimization sequence to extract $w^\star$. The right panel of Figure 4 presents different measurements for epoch budgets ranging from 100 to 280, with all other parameters kept consistent. Our findings indicate a relative similarity in results before the sequence nears $w^\star$, suggesting that our experiments do not display excessive sensitivity to the epoch budget.

• **Induced Bias:** However, this experiment also underscores the phenomenon detailed in Section 4.1: as the sequence approaches completion, the correlation between sampled gradients and $w_t - w^\star$ - induced by gradient updates - becomes increasingly significant. This correlation is a by-product of the optimization method, rather than a feature of local geometry, and augments the value of RSI and $\gamma$ by diminishing the impact of stochasticity. Consequently, this correlation should be taken into account when interpreting RSI and EB in the concluding epochs.

A compelling illustration of this correlation can be seen in a discrete isotropic random walk with a fixed step size $s$ in a dimension $d$. When dimension $d$ significantly exceeds the number of steps, each pair of steps can be assumed to be nearly orthogonal with high probability. In such a setting, if we denote $(x_t)_{t=0\dots T}$ as the sequence generated by the random walk, we can calculate that, with high probability, $\forall t$,

$$\frac{(x_t - x_{t+1})^T (x_t - x_T)}{\|x_t - x_T\|_2^2} \approx \frac{\|x_t - x_{t+1}\|_2^2}{\|x_t - x_T\|_2^2} \approx \frac{1}{T-t} > 0 \quad \text{and} \quad \frac{\|x_t - x_{t+1}\|_2}{\|x_t - x_T\|_2} \approx \frac{1}{\sqrt{T-t}} \quad (8)$$

Consequently, the cosine similarity $\gamma(x_t) \approx (T-t)^{-0.5}$ remains strictly positive, and experiences a sharp increase toward the end, exemplifying the effect of the correlation induced by the selection of $w^\star$. It's worth noting that in the case of neural networks, $\gamma$ remains approximately constant for the majority of training (as is clearly visible in Figure 1), which marks a distinction in their dynamics. Nonetheless, akin to the random walk scenario, it can be anticipated that the correlation induced by the choice of $w^\star$ would become increasingly evident as the number of remaining iterations diminishes.

**Contrasting Examples: Functions Without Beneficial Geometric Properties:** We now turn our attention to delineating the behaviors that could potentially manifest in stochastic and non-convex optimization scenarios. To this end, we have engineered two illustrative counter-examples which effectively demonstrate that the consistency observed in Sections 4 and 5 is not a mere byproduct of our experimental paradigm. Our first example, termed Asymmetric Linear Model (ALM), entails the training of a linear model with the objective of consistently yielding outputs that are lower than their corresponding targets. The error between these values is calculated on stochastic minibatches using

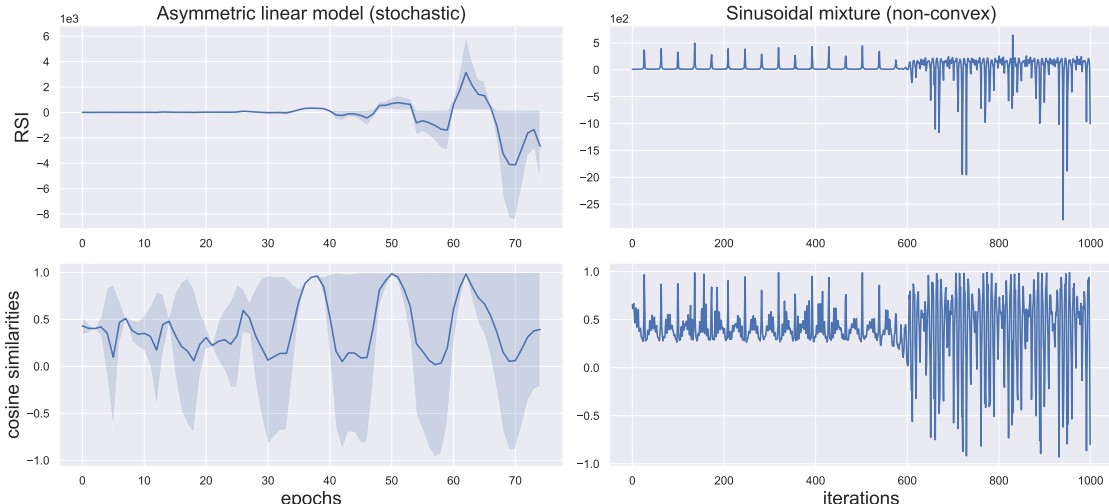

Figure 5: Left column: RSI and $\gamma$ values for a convex yet significantly stochastic objective. Right column: RSI and $\gamma$ values for a deterministic objective characterized by substantial non-convexity. Despite their simplicity, these functions exemplify the irregular behaviors one might anticipate encountering within the complex terrain of neural loss landscapes.

Root Mean Square Error (RMSE), thereby introducing a substantial degree of stochasticity. Despite this, the objective is a finite sum of convex functions and thus remains convex.

The second function, designated Sinusoidal Mixture (SM), is deterministic but exhibits a pronounced degree of non-convexity. The mathematical expressions for both ALM and SM are presented below, with coefficients $a_i, x_i, y_i$ drawn randomly from normal distributions,

$$ALM(w) = \sum_i \left[ max(0, w^T x_i - y_i) \right]^2 ; \quad SM(w) = \|w\|_2^2 + 100 \sum_i \sin(a_i w_i)^2. \quad (9)$$

Figure 5 presents the measurements of RSI and $\gamma$ for both ALM and SM. Although these functions are characterized by relatively simple functional forms and do not simultaneously exhibit stochasticity and non-convexity, they demonstrate unpredictable trajectories and negative values for RSI and $\gamma$. This evidence compellingly suggests that the observed simplicity associated with neural networks is not a trivial characteristic.

## 7 CONCLUSION

We have conducted an extensive series of experiments, assessing RSI and EB across a broad spectrum of training settings. These experiments reveal that these geometric properties display a collection of desirable characteristics, effectively demonstrating that neural network training proceeds smoothly, maintaining a consistently steady advancement towards its destination throughout the training process.

These results contrast starkly with the theoretical complexity of neural landscapes and potentially open new pathways for developing theoretical results tailored to deep learning, or for designing optimization algorithms that exploit the geometry of empirical objective functions.

A noteworthy point is that while RSI and EB appear to encapsulate significant beneficial aspects of neural networks, they likely do not encompass the entire scope of these advantages. There may be additional, complementary properties yet to be discovered. An intriguing indication of this is the fact that vanilla gradient descent has been proven to be exactly optimal for functions verifying the lower restricted secant inequality and upper error bound (Guille-Escuret et al., 2022). Given the well-documented efficacy of momentum in training neural networks, we conjecture that momentum exploits additional properties not captured by RSI and EB, which we encourage future works to explore.

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

## A EXPERIMENTAL SETTING

### A.1 ALGORITHM

Algorithm A.1 provides a detailed account of our experimental protocol. This algorithm describes our methodology to measure RSI and EB values throughout the training process.

Intuitively, the algorithm follows two training runs with identical initialization and minibatch sampling. The first runs aims at computing the last iterate $w^\star$ and saving it, and the second runs uses $w^\star$ to compute RSI and EB.

This approach removes the necessity to save all gradients throughout the run (in order to compute RSI and EB at the end), which would be unreasonably expensive in memory.

---

**Algorithm 1** Measurement of RSI and EB

**Input:** initial weights $w_0$, sequence of minibatches $\mathcal{B}_{0...T-1}$

1: **for** $t = 0, \ldots, T - 1$ **do**
2:     compute gradient $G_t = \nabla \mathcal{L}_{\mathcal{B}_t}(w_t)$
3:     update weights $w_{t+1} = Opt(w_{0...t}, G_{0...t}))$
4: **end for**
5: $w^\star \leftarrow w_T$
6: reset weights to $w_0$
7: **for** $t = 0, \ldots, T - 1$ **do**
8:     compute gradient $G_t = \nabla \mathcal{L}_{\mathcal{B}_t}(w_t)$
9:     compute $RSI_t = \frac{G_t^T(w_t - w^\star)}{\|w_t - w^\star\|_2^2}$
10:     compute $EB_t = \frac{\|G_t\|_2}{\|w_t - w^\star\|_2}$
11:     update weights $w_{t+1} = Opt(w_{0...t}, G_{0...t}))$
12: **end for**

**Output:** $RSI_{0...T-1}, EB_{0...T-1}$

---

### A.2 IMPLEMENTATION AND ENVIRONMENT FOR EXPERIMENTS

**Computational Environment**: We perform our experiments mainly with cluster A (redacted until publication). For cluster A, each node is composed of NVIDIA A100×4GPU and AMD Milan 7413 @ 2.65 GHz 128M cache L3×2CPU. As a software environment, we use Rocky Linux 8.7, gcc 9.3.0, Python 3.10.2, pytorch 1.13.1, torchvision 0.14.1, cuDNN 8.2.0, and CUDA 11.4.

**Licence of Datasets**: It should be noted that the CIFAR-10 dataset (Krizhevsky et al., 2012) does not explicitly stipulate any licensing terms [1]. The authors of the CIFAR-10 merely ask users of their dataset to provide appropriate citation. ImageNet-1K (Deng et al., 2009) does not explicitly state its license [2]. Licenses of the WikiText-2 (Logan et al., 2019) is CC-BY-SA-3.0 [3]. No license is specified for the dataset in Vaihingen (Cramer & Haala, 2010), but it is allowed to be used in scientific papers. However, acknowledgment and citation are required[4].

**Implementation**: All codes for experiments are modifications of the codes provided by PyTorch's official implementation for image classification and language modeling tasks[5] and Audebert et al. (2017) for segmentation task [6]. The license for the official Pytorch implementation is the BSD-3-Clause, and the license for the segmentation task implementation is GPLv3. Our code can be found at the link below.
`https://github.com/Hiroki11x/LossLandscapeGeometry`

---

[1] `https://www.cs.toronto.edu/~kriz/cifar.html`
[2] `https://www.image-net.org/challenges/LSVRC/2012/index.php`
[3] `https://www.salesforce.com/products/einstein/ai-research/the-wikitext-dependency-language-modeling-dataset/`
[4] For more details, see page 7 of `https://www2.isprs.org/media/komfssn5/complexscenes_revision_v4.pdf`
[5] `https://github.com/pytorch/examples`
[6] `https://github.com/nshaud/DeepNetsForEO`

## A.3 Datasets description

**CIFAR10**: CIFAR-10 dataset (Krizhevsky et al., 2012), one of the most widely used datasets for machine learning research, is a unique resource that offers a robust benchmark for algorithms, primarily image recognition. The dataset is a curated collection of 60,000 color images, each of a size of 32x32 pixels, uniformly divided across ten distinctive classes. These classes encompass various common objects: airplanes, automobiles, birds, cats, deer, dogs, frogs, horses, ships, and trucks. Each class in the CIFAR-10 dataset is represented equally, with 6,000 images per category. The dataset is split into two segments: a training set comprising 50,000 images and a test set of 10,000 images.

**ImageNet-1K**: The ImageNet-1K dataset, a subset of the more extensive ImageNet database (Deng et al., 2009), has become an essential resource for research in machine learning, particularly for image recognition and classification tasks. ImageNet-1K is an extensively curated dataset of approximately 1.28 million high-resolution color images spread across 1,000 distinct categories or classes. These classes span various objects, organisms, and phenomena, capturing a rich diversity of the visual world.

**WikiText-2**: The WikiText-2 dataset (Logan et al., 2019) is a significant benchmark for various natural language processing tasks, specifically those related to language modeling. It comprises over 2 million tokens extracted from verified Wikipedia articles. WikiText-2 retains the original structure and complexity of the language found in the source articles. This characteristic has enabled training models to handle various language structures and styles. The dataset is divided into three segments: a training set with roughly 2.08 million tokens, a validation set with approximately 217,000 tokens, and a test set with about 245,000 tokens.

**Vaihingen**: The Vaihingen dataset (Rottensteiner et al., 2012) is a land covering remote sensing dataset. Its purpose is to segment correctly aerial images of the Vaihingen city in Germany. It is composed of 33 tiles and we use 11 tiles for training, 5 tiles for validation, and the remaining 17 tiles for testing our model, which is the split used in (Fatras et al., 2021). Furthermore, we only consider the RGB components of the Vaihingen dataset. We follow the training procedure and PyTorch implementation from (Audebert et al., 2017). We build our training (resp. validation) dataset by taking randomly $256 \times 256$ patches from the training (resp. validation) tiles. The number of images seen during a training epoch is set to 10.000 patches while it is set to 1000 for the validation set.

Table 1: Default setting of experiments

| Task | Dataset | Model | Batch size | Epochs |
|---|---|---|---|---|
| Image Classification | CIFAR-10 | ResNet18-1 | [64, 128, 256, 512] | [100, 190, 280] |
| | | Medium-MLP | [64, 128, 256, 512] | [100, 190, 280] |
| | ImageNet-1K | ResNet-18-1 | 256 | [90, 180] |
| | | ResNet-18-2 | 256 | [90, 180] |
| | | ResNet-50-1 | 256 | [90, 180] |
| | | ResNet-50-2 | 256 | [90, 180] |
| | | ResNet-152-0.5 | 256 | [90, 180] |
| | | ResNet-152-1 | 256 | [90, 180] |
| Word Language Model | WikiText-2 | Transformer | [32, 64, 128, 256] | 20 |
| Segmentation | Vaihingen | UNet | 10 | 26 |
| | | SegNet | 10 | 26 |

## A.4 Hyperparameters and Detailed Configurations

In the experimental procedure of our study, we employed a systematic grid search method to explore hyperparameters. This approach facilitates the identification of the most effective combinations that provide superior performance.

The specifics concerning the batch size and the total number of epochs allocated for each dataset and corresponding model have been exhaustively tabulated in Table 1. These parameters were meticulously selected to ensure optimal learning while mitigating overfitting concerns.

Further, we present detailed settings of specific ablation experiments in Table 2, 3,4, and 5. These ranges were defined based on prior search of hyperparameters maximizing validation performance.

'SGD' denotes the standard SGD algorithm without momentum, 'Momentum' denotes SGD with momentum with $\beta = 0.9$, and 'Adam' denotes the Adam algorithm with $\beta_1 = 0.9$ and $\beta_2 = 0.999$.

**CIFAR10**: We train a ResNet-18 (He et al., 2016) for 190 epochs on CIFAR-10 (Krizhevsky et al., 2012) with SGD + momentum using a batch size of 256, a weight decay of $10^{-6}$, and a fixed step size of $10^{-2}$ as a default configuration.

For batch-size experiments, the learning rate was designated as $5.0 \times 10^{-3}$ for a batch size of 64, and subsequently scaled proportionally to the square root of the batch size, adhering to the guidelines from prior research (Krizhevsky, 2014).

**ImageNet-1K**: We train a ResNet-50 on ImageNet Deng et al. (2009) for 180 epochs with SGD + momentum using a batch size of 256, weight decay of $10^{-4}$, and a learning rate of $10^{-3}$. The learning rate is subjected to a linear warmup for the first 3 epochs, followed by cosine annealing as a default configuration. We indicate by 'max LR' the maximum value of the learning rate, reached after the warmup epochs.

**WikiText-2**: We train a transformer Vaswani et al. (2017) [7] on WikiText-2 Merity et al. (2016) for 20 epochs with Adam Kingma & Ba (2017) using a batch size of 32, weight decay of $10^{-5}$ and learning rate of $10^{-4}$.

**Vaihingen**: We train a SegNet Badrinarayanan et al. (2017) and a UNet (Ronneberger et al., 2015). We augment our data with flip and mirror transformations. We use a batch size of 10 patches taken randomly within images as done in Audebert et al. (2017). We train for 25 epochs with SGD + momentum, learning rate 0.01 and weight decay $1e^{-5}$. We then do an extra epoch with the learning rate and the weight decay divided by 10. Note that we train both the UNet and the SegNet from *scratch*.

Table 2: Hyperparameter: Image Classification Task (CIFAR-10)

| Task | Model | Dataset | Optimizer | Batch size | LR | Epochs Budget |
|------|-------|---------|-----------|------------|-----|---------------|
| Optimizer | ResNet18-1 | CIFAR-10 | [SGD, Momentum, Adam] | 256 | [0.0001, 0.0005, 0.001] | [100, 190, 280] |
| Seed | ResNet18-1 | CIFAR-10 | Momentum | 256 | 0.01 | 190 |
| Batch Size | ResNet18-1 | CIFAR-10 | Momentum | [64, 128, 256, 512] | 0.005 [8] | 190 |
| Model | [Medium-MLP, ResNet18-2] | CIFAR-10 | Momentum | 256 | 0.01 | 150 |

Table 3: Hyperparameter: Image Classification Task (ImageNet-1K)

| Task | Model | Dataset | Optimizer | Batch size | max LR | Epochs Budget |
|------|-------|---------|-----------|------------|--------|---------------|
| Model | [ResNet18-1, ResNet18-2, ResNet50-1, ResNet50-2, ResNet152-05, ResNet152-1] | ImageNet-1K | Momentum | 256 | 0.1 | [90, 180] |

Table 4: Hyperparameter: Language Model Task (WikiText-2)

| Task | Model | Dataset | Optimizer | Batch size | LR | Epochs Budget |
|------|-------|---------|-----------|------------|-----|---------------|
| Batch Size | Transformer | WikiText-2 | Adam | [32, 64, 128, 256] | 0.0001 [9] | 20 |

---

[7] We use pytorch official implementation of transformer for language model: `https://github.com/pytorch/examples/blob/main/word_language_model/model.py`

[8] The base learning rate is configured with an assumption of a batch size of 64. If the batch size is doubled, the learning rate should be multiplied by the square root of 2.

[8] Same as above.

[9] Same as above.

Table 5: Hyperparameter: Segmentation Task (Vaihingen)

| Task | Model | Dataset | Optimizer | Batch size | LR | Epochs Budget |
|------|-------|---------|-----------|------------|------|---------------|
| Model | [UNet, SegNet] | Vaihingen | Momentum | 10 | 0.01 | 26 |

## A.5 VALIDATION PERFORMANCE

To support the relevance of our experimental setting, we report the validation performance in the standard settings of each dataset and model.

Table 6: Validation accuracy on CIFAR-10 with batch size 256.

| Model | Validation accuracy |
|-------|---------------------|
| ResNet18-1 | 90.25 |
| Vanilla MLP | 59.42 |

Table 7: Validation accuracy on ImageNet with batch size 256.

| Model | Validation accuracy |
|-------|---------------------|
| ResNet18-1 | 67.63 |
| ResNet18-2 | 69.75 |
| ResNet50-1 | 72.31 |
| ResNet50-2 | 73.67 |
| ResNet152-0.5 | 72.23 |
| ResNet152-1 | 73.07 |

Table 8: Validation perplexity on WikiText-2 with batch size 64.

| Model | Validation perplexity |
|-------|-----------------------|
| Transformer | 60.72 |

Table 9: Validation accuracy on Vaihingen with batch size 10.

| Model | Validation accuracy |
|-------|---------------------|
| SegNet | 84.56 |
| UNet | 85.40 |

## B ABLATION STUDY

In this section, we provide additional results that did not fit in the main paper for ablation studies. For instance, in addition to the cosine similarities presented in figure 1, we provide the individual values of RSI and EB.

We provide in Figure 6 the cosine similarities for different batch sizes on CIFAR-10, as a complement to Figure 1 to study the impact of batch size RSI and EB.

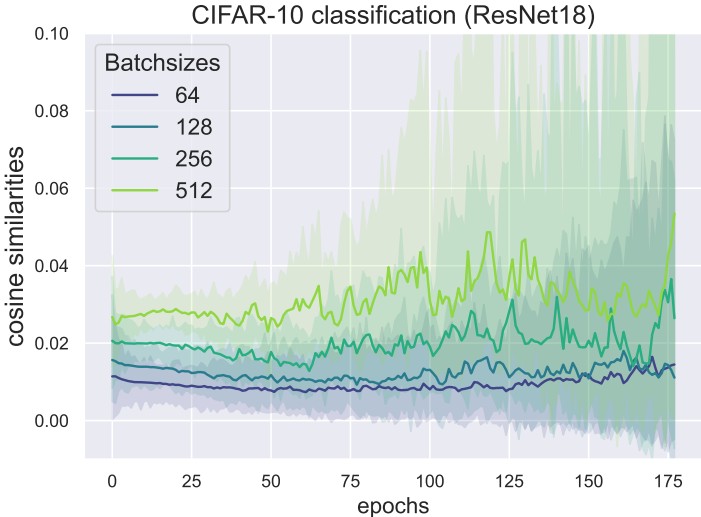

Figure 6: cosine similarities measured during the training of a ResNet-18 on CIFAR-10, for batchsizes ranging from $64$ to $512$.

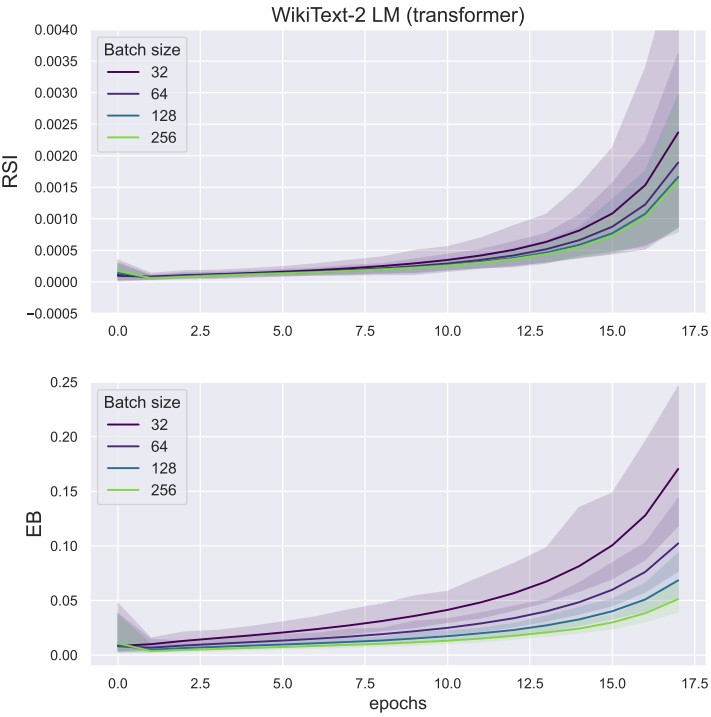

Figure 7: RSI and EB throughout training of a transformer on WikiText-2 with different batch sizes. This figure is complementary to figure 1.

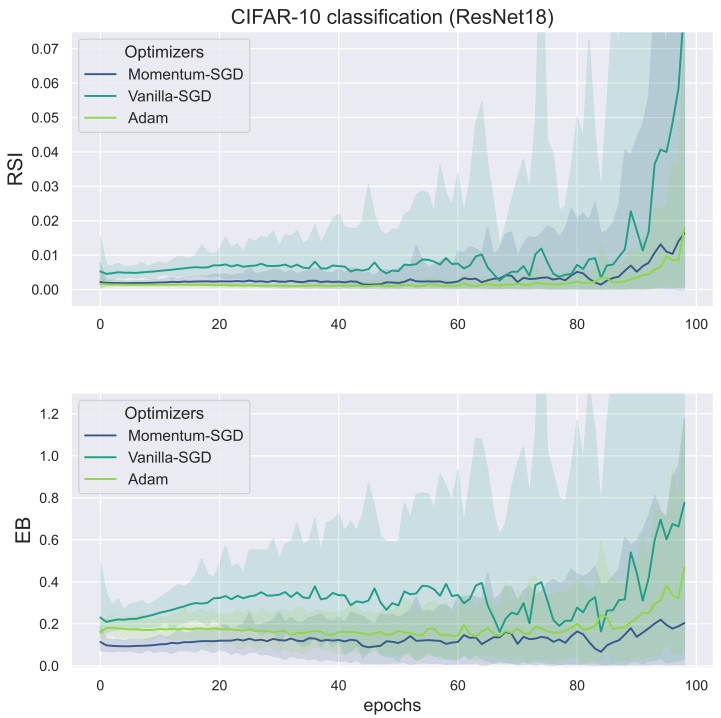

Figure 8: RSI and EB throughout training for the training of a ResNet18 on CIFAR-10 with different optimizers. This figure is complementary to figure 1.

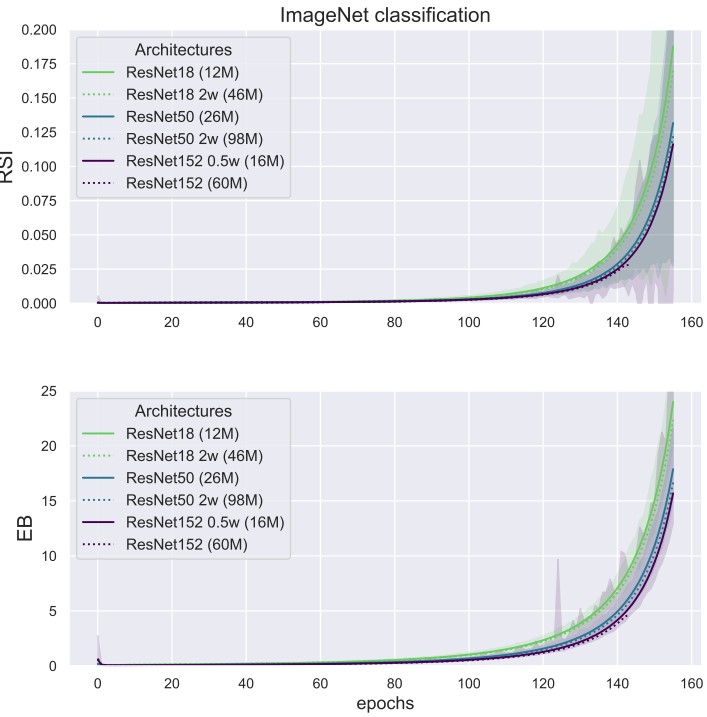

Figure 9: RSI and EB throughout training for the training of different ResNet architectures on ImageNet. This figure is complementary to figure 1.

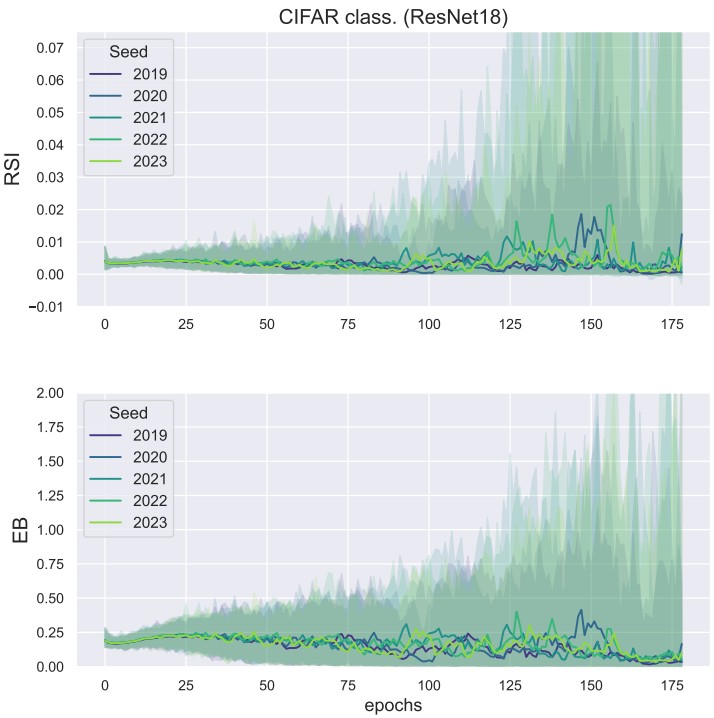

Figure 10: RSI and EB throughout training for the training of a ResNet18 on ImageNet with different random seed. This figure is complementary to figure 4.

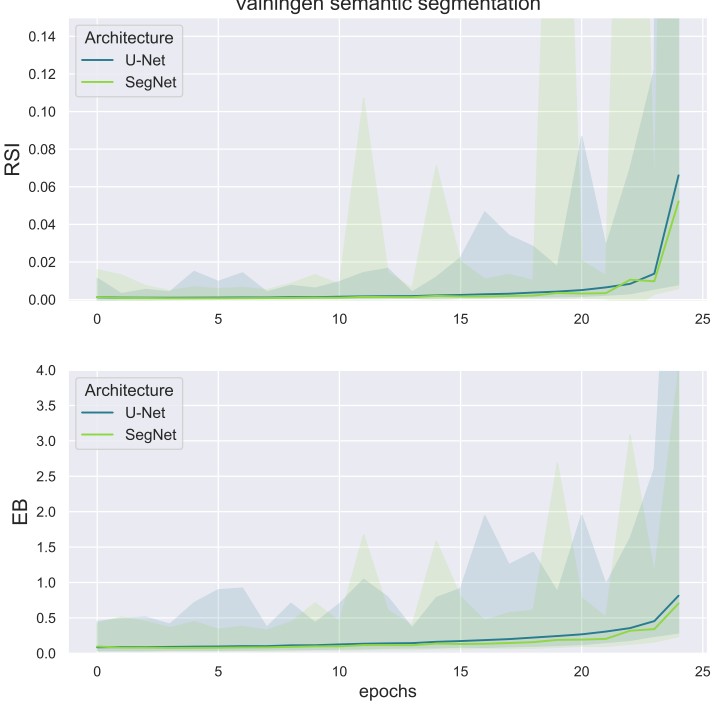

Figure 11: RSI and EB throughout training for two different architectures on Vaihingen semantic segmentation. This figure is complementary to figure 1.

## C    ADDITIONAL FIGURES

In this section we introduce additional figure supporting claims or conjectures made in the main paper.

Figure 12 shows the evolution of $\|w_t - w^\star\|_2$ throughout training. We can see Adam traverses a larger distance than Vanilla SGD and Momentum SGD, and evolves as a more regular pace. We believe this could be a factor in the lower cosine similarities exhibited by Adam in Figure 1.

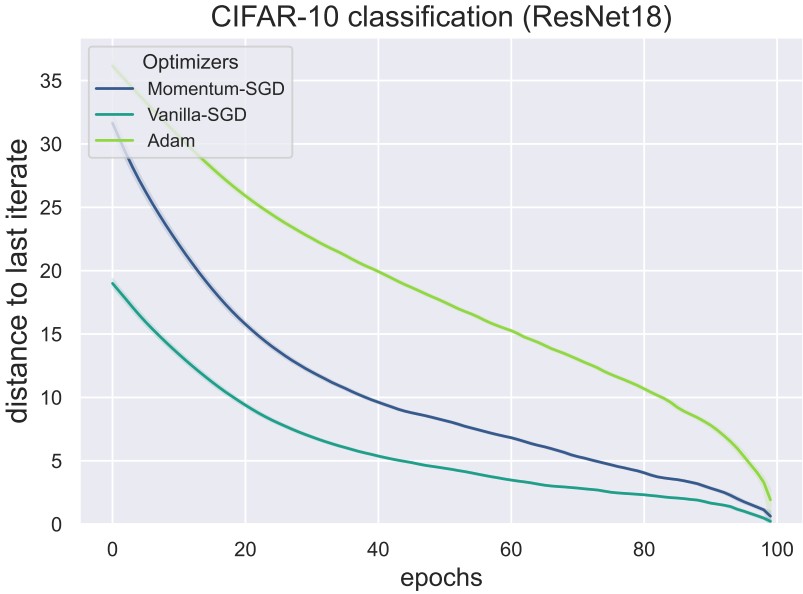

Figure 12: $\|w_t - w^\star\|_2$ over training of a ResNet18-1 on CIFAR-10, with different optimizers.

Figure 13 indicates the value of $\|w_t - w^\star\|_2$ over training in the three settings of Figure 2. An important remark is that due to the cosine decreasing learning rate schedule, in the case of ImageNet, this distance becomes negligible in the last 25 epochs. This raise precision issues as discussed in section 4.1. Since $w_t$ is subject to negligible variations in the last 25 epochs of the ImageNet experiment, Figures 1 and Figure 2 omit the epochs after 155 in the case of ImageNet, in order to improve readability and focus on meaningful settings.

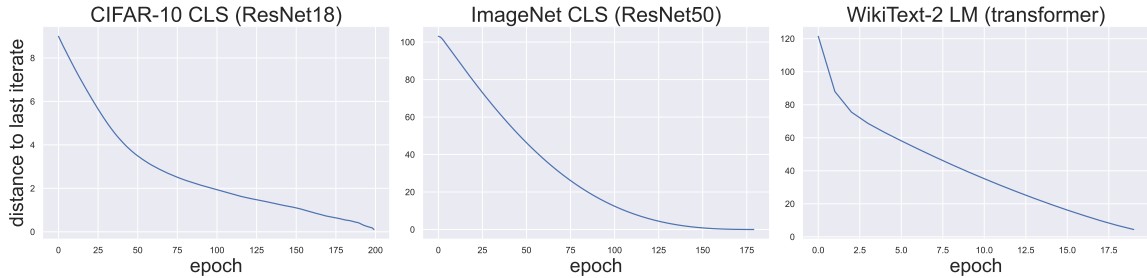

Figure 13: $\|w_t - w^\star\|_2$ over training in the three different settings of Figure 2.

# D  PLAUSIBLE CAUSES

In this section, we examine potential factors that might be contributing to the remarkable geometric regularity observed via RSI and EB within the loss landscapes of neural networks. These are conjectural in nature, and we advocate for more rigorous investigation in future work to substantiate these propositions.

• **Architectural Characteristics:** The deep learning landscape has witnessed a plethora of architectural enhancements since its inception. Notably, ResNets incorporate advanced features such as skip-connections and batch normalization (Ioffe & Szegedy, 2015b), which were found to simplify the structure of the loss landscape (Santurkar et al., 2018a; Li et al., 2018). It is plausible that such favorable geometric attributes could play a significant role in the success of neural network architectures. Therefore, our observations may be more a byproduct of the selection of high-performing networks rather than an universal characteristic.

Nonetheless, Figure 14 provides an interesting comparison of cosine similarities derived from the training of a wide 4-layer Perceptron (MLP) with ReLU activations and a double-width ResNet-18. Despite a similar parameter count, these two architectures exhibit a considerable performance gap. Intriguingly, not only does the MLP not exhibit inferior geometrical properties, but it actually shows greater regularity in cosine similarity compared to the ResNet-18. This result suggests that the beneficial geometry of neural loss landscapes is not simply a consequence of extensive architectural tuning, but potentially a more intrinsic feature. Nonetheless, prior work concluded that skip connections significantly simplify the loss landscape at higher depth (Li et al., 2018).

• **High Dimensionality:** Our conjecture is that the primary contributor to the regular patterns observed by RSI, EB, and $\gamma$ is the large dimensionality of neural loss landscapes. Assuming that a significant number of dimensions maintain a degree of independence, even when the gradient occasionally points in the 'wrong direction' in certain dimensions, this can be offset by averaging over a sufficiently vast number of dimensions. However, formalizing such an effect is challenging due to the evident dependencies between dimensions.

• **Properties of Real-World Data:** Lastly, the geometric simplicity of optimization paths might be influenced by inherent properties of real-world data distributions. For instance, it is commonly postulated that unstructured data from real-world applications resides within lower-dimensional manifolds. Analogous properties could be pivotal in shaping loss landscapes, which appear more benign than what worst-case scenarios might suggest.

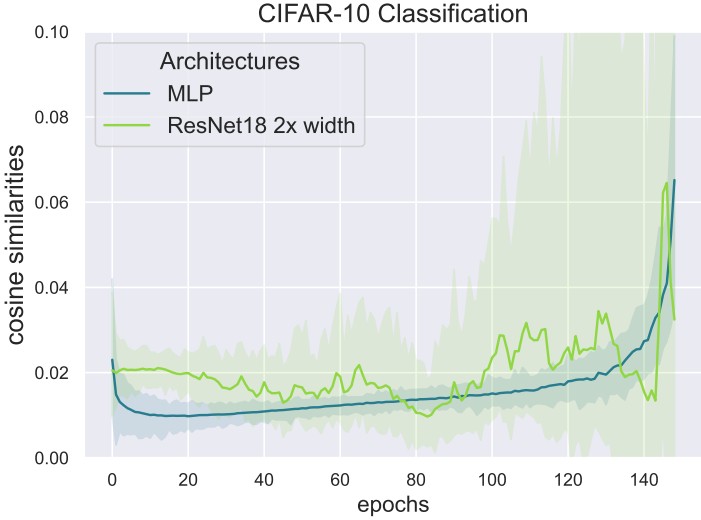

Figure 14: Comparison of cosine similarities derived from a CIFAR-10 classification task employing two distinct architectures: a straightforward Multi-Layer Perceptron (MLP) and a more complex ResNet-18 structure.

