# OpenReview forum: "No Wrong Turns: The Simple Geometry Of Neural Networks Optimization Paths"
_ICLR.cc/2024/Conference — Submitted to ICLR 2024_

### Official Review · Reviewer_Km89 · 2023-10-26

**Soundness:** 2 fair
**Presentation:** 3 good
**Contribution:** 2 fair
**Rating:** 3
**Confidence:** 4

**Summary:**

This paper proposes several measurements to quantify the optimization dynamics of training neural networks. Motivated by optimization theory, the authors study how cosine similarity, Lipschitz-type, and convexity-type quantities evolve over the training process.

By empirically studying those quantities on many datasets, ranging from images to languages, the authors claim that the optimization geometry is stable and simple, and in particular "optimization trajectories never encounter significant obstacles". Based on these observations, the authors provide insights and discussions about common training techniques such as initialization and batch size.

**Strengths:**

1. Overall the observation that optimization dynamics is benign for state-of-the-art neural networks is informative and interesting. This reinforces prior observations such as those in Goodfellow et al. (2015).

2. Empirical measurements such as RSI, EB, and cosine similarity are useful diagnostic tools for understanding the optimization property of neural networks

**Weaknesses:**

Overall, I find the paper seems to be overclaiming the results.

1. (main) This paper seems to give a misleading message. The title says the neural networks have "simple geometry", and the abstract says "optimization trajectories never encounter significant obstacles, [....] maintain stable dynamics during the majority of training." I think that those properties are examined only for state-of-the-art neural networks. There are lots of techniques to alleviate optimization issues, so the overall claim of this paper severely trivializes the techniques in common practice. (On a related note: It would be interesting to investigate how much each technique contributes to making optimization benign.)

2. (main) I find that RSI is confusing to understand: in Figure 2, the measure for ImageNet and WikiText is initially close to zero, so does this mean the training dynamics is not ideal in the beginning? "RSI and EB follow predictable trends" do not imply that optimization dynamics have benign properties.

3. (main) The cosine similarity is low. While I agree with the argument that there is stochasticity, I think cosine similarity should be compared with the baseline where (i) gradients are replaced by noise, or (ii) gradients are calculated based on random labels or random tokens in a minibatch. Without comparison with the pure noise baseline, we do not know if cosine similarity is significantly correlated with the signal direction.

4. (minor) I understand that this paper is mainly about investigating the empirical properties of training neural networks from the optimization perspective. But from a practical point of view, what can we learn? Do we know what part of the architecture of, say a transformer, is crucial for benign optimization? Which techniques (layer normalization, dropout, learning rate warmup, etc) are playing a role?

**Questions:**

Please see the weaknesses section.

---

> ### Author Response · Authors · 2023-11-21
> **Rebuttal**
>
> We thank the referee for their careful consideration and feedback.
>
> **Weakness 1**
> We agree with the reviewer that architectural improvements are likely a big contributor to the phenomenon we observe. In fact, in Appendix D, we list architectural characteristics as a possible factor to the behavior we observe, and discuss it (see architectural characteristics paragraph). While we observe that the stable behavior of cosine similarity is, in fact, preserved on simple architectures (vanilla MLP with ReLU activations), our experiments cannot conclude on the contribution of e.g. batch norm, skip connections, etc.
> We agree this is an interesting line of research for future work. However, as our experiments are quite expensive (requires double computation as each training run needs to be computed twice), the range of ablations we have conducted is already quite extensive.
>
> Most importantly, the main goal of our work is to characterize the properties of neural loss landscapes in practical settings, i.e. using modern architectures. As we discuss, our main goal is not to explain why these properties are present, but to quantify them and their relation with various factors (such as optimizers, depth and width, seeds, task, etc...). Since these properties seem very relevant to training dynamics, better understanding them mean we can better understand training behavior, and build a more practically relevant theory. Thus, while indeed our observations are only validated on heavily engineered architectures, we argue that this instead makes our work more relevant, and validating our claims on simple architectures that are not deployed in practice would be less meaningful.
>
> **Weakness 2**
> The values of RSI and EB appear close to $0$ in Figure 2 due to being small compared to the late stages of training, where both RSI and EB go to infinity (see late training behavior paragraph). But this is just a matter of scale, and the cosine similarity in fact quickly stabilizes.  The speed at which the iterates $w_t$ approach $w_T$ is in fact entirely determined by RSI, EB, and the learning rate (see Equation 4), and the value of RSI and EB in the early stages, while small in comparaison to late training, are not negligible. For a fixed cosine similarity $\frac{RSI}{EB}$, observing smaller values of RSI and EB simply means the ideal learning rate should be larger.
> "RSI and EB follow predictable trends" does mean that the optimization trajectory advances toward its last convergence point at a steady pace, and that despite non-convexity and stochasticity, all iterations (with very few exceptions) contribute similarly in term of parameter space progression. The stability of the values of RSI and EB across minibatches, and its predictability across phases of training, mean that ideal values of the LR are stable across minibatches, and predictable across phases of training. This explains why simple LR schedules can be so efficient in training neural networks despite their non-convexity and the stochasticity of gradient sampling.
>
> **Weakness 3**
> We do compare to the baseline where gradients are replaced by noise, as suggested by the referee, in Section 6 (paragraph "induced Bias"). We also discuss the low value of cosine similarity in Section 4 (paragraph "low value of cosine similarity") and remark that in fact, if it wasn't very low, we would be able to train neural networks in very few steps, which know not to be the case. The cosine similarity being low can be compared to a quadratic function being badly conditioned: optimization is difficult, but follows a simple underlying structure, which eventually drives the convergence. Interestingly, RSI and EB would also follow predictable trends in the case of a random walk due to the high dimensionality having a regularizing effect. This can be interpreted as random walks having a stationary update rule. The complex structure of neural networks suggest the training dynamics could be highly complex (see contrasting examples in Section 6). Our work finds that it is not the case.
>
> **Weakness 4**
> As stated in our reply to Weakness 1, our work does not provide direct insights on which architectural features enable such properties. The goal of our work is to improve our understanding of the fundamental properties of neural loss landscapes that make first order optimization so surprisingly efficient, rather than exploring their origin. Once a sufficiently refined characterization of empirical neural loss landscapes is obtained, we will be able to derive theoretical results in more realistic settings than previous works, which will hopefully be able to guide the design of more efficient optimization algorithms. A few more steps in this direction are therefore necessary to lead to immediate algorithmic improvements.

---

### Official Review · Reviewer_LuUd · 2023-11-01

**Soundness:** 3 good
**Presentation:** 4 excellent
**Contribution:** 3 good
**Rating:** 6
**Confidence:** 3

**Summary:**

This paper studies the geometric properties of NN training loss over the training procedure. The authors consider two conditions called Lower Restricted Secant Inequality (RSI) and Upper Error Bounds (EB). The paper tracks the local and stochastic versions $RSI(G,w,w^*)$ and $EB(G,w,w^*)$ as well as their ratio $\gamma(G,w,w^*) = \frac{RSI(G,w,w^*)}{EB(G,w,w^*)}$, which are quantities evaluated with the stochastic gradient oracle $G$ queried for the current point $w$. In particular, $\gamma$ corresponds to the cosine similarity of the stochastic gradient and $w-w^*$, so $\gamma > 0$ implies that the stochastic gradient is pointing to the "right" direction. For various settings, the paper demonstrates that the RSI, EB, and $\gamma$ stay quite stable throughout training, despite the nonconvexity and stochasticity. Moreover, the cosine similarity $\gamma$ is almost always positive. The paper supplements the main results with ablations and detailed discussions.

**Strengths:**

1. I think this paper is a good submission. It provides an interesting set of results that sheds light on the theory-practice gap in neural network training. The empirical observations suggest that neural network training, although nonconvex and stochastic, could be in fact much more well-behaved than expected.

2. The paper is well-written and delivers the main ideas clearly. The ablation studies and discussions at the end of the paper provide useful insights.

3. The analysis of locally optimal learning rate seems to have a potential to lead to a method for learning rate schedule design or for adaptive learning rate schemes.

**Weaknesses:**

1. There is a closely relevant paper titled "SGD Converges to Global Minimum in Deep Learning via Star-convex Path" which appeared in ICLR 2019, which also empirically studies the SGD trajectories and shows that the star-convexity is satisfied for most epochs. I have not checked the follow-up results carefully, so there may be more existing results that discover surprisingly benign geometric properties of neural network training. The paper should contextualize itself relative to this existing result(s).

**Questions:**

1. A recently discovered phenomenon Edge of Stability (EoS) shows that GD trajectory experiences progressive sharpening and then reaches and hovers around the stable convergence threshold $2/\text{(step size)}$. When EoS is in effect in the later phase of training, the training dynamics is rather unstable, with occasional peaks in loss curves. Similarly, when you run SGD instead of GD, we also often observe these loss peaks in the training curves. In contrast, the RSI, EB, and $\gamma$ curves look quite stable, without any significant peaks or perturbations. My question is: did you observe any peaks or unstable convergence in your experiments? Do the "fundamental properties" (Section 4.1) continue to hold when we observe EoS? If so, how can we reconcile these seemingly contradictory trends?

2. In the late phase of training, the curves seem to experience sharp increases in both mean and variance. The explanations based on interpolation vs non-interpolation and the bias introduced by setting $w^* \approx w_T$ are intuitive, but do they explain the whole story? For example, in Figure 2, the RSI and EB of CIFAR-10 do not show a significant increase in mean curves, but we do observe a consistent increase in variance. What could be an explanation for this?

3. It seems that using large batch sizes increases $\gamma$ throughout training. What happens if we use full-batch training (i.e., GD)?

---

> ### Author Response · Authors · 2023-11-20
> **Rebuttal**
>
> We thank the referee for their careful consideration and feedback.
>
> **Weakness 1:**
>
> This is an excellent point. We were surprised to find that "SGD Converges to Global Minimum in Deep Learning via Star-convex Path" [1] is not already part of our bibliography, as it is a paper that has inspired our work and was one of the first added to our bib file. We intended to discuss differences with this seminal work in detail, and to be transparent, we forgot. We will add a paragraph contextualizing our work with respect to this paper.
>
> The limitations in [1] that we try to address in ours are the following:
> - Their work assume interpolation regime and only evaluates on CIFAR and MNIST, without typical training components such as batch normalization, weight decay, momentum, etc. By evaluating in realistic settings, with popular architectures and optimizers, on a variety of more advanced tasks (including in the non-interpolation regime), our work is closer to empirical practice.
> - The epoch-wise star-convexity is fairly limited as it only considers the average of first-order developments over a full epoch. The iteration-wise star-convexity is only verified with respect to the last iterate of the sequence (just like in our work) but their theoretical results require star-convexity with respect to *all* minimizers of the given minibatch, which seems impossible to confirm empirically (and is in fact unlikely due to permutation invariance). Our theoretical results do not have such limitations and apply directly from our observations.
> - [1] Assumes L-smoothness of each $l_i$. Even when the loss is made smooth, it typically entails an extremely high constant $L$ which is unlikely to explain the good behavior of training, and it is intractable to estimate in practice. In our work, the role of smoothness is played by EB.
> - Perhaps most importantly, our work **quantifies** the properties of optimization paths on neural networks. In [1], the paths are simply observed to be star-convex, which would be akin to only observing the non-negativity of RSI (or $\gamma$) in our work. This quantification allows us to suggest LR schedules, guarantee linear convergence, etc.
>
> In summary, prior works ([1], Goodfellow et al. (2015) and Lucas et al. (2021)) observed that the loss landscapes is simpler than expected, whereas we attempt to formally characterize and quantify this simplicity, providing more advanced tools to build theoretical results, and making a significant step toward practical relevance in our opinion.
>
> **Question 1 :** since we train in realistic settings, we inevitably observe jumps in the training loss. The reason this does not contradict EoS observations is that small perturbations in high-curvature directions (like a small modification of very impactful features) can lead to large variations of the training loss, without impacting significantly the trajectory in the parameter space.
>
> **Question 2 :** We believe this is due to numerical issues due to the gradients and $||w-w_T||_2$ becoming very small, and the fact that not all samples are interpolated as closely, and a few samples are not well interpolated at all. While this is insignificant in term of average training loss, and not noticeable before convergence, when nearing convergence this induces significant variance in RSI and EB. Indeed, minibatches containing samples that are not well interpolated will suffer form the same fate as training in the non-interpolation regime. While these samples are rare, since the shadowed area correspond to the minimum and maximum values over *all* steps in a epoch, this effect can become signifiant.
>
> **Question 3:** We assume RSI and EB would be even better behaved by taking out the stochasticity. We did not conduct these experiments, first because it would be very computationally expensive to train until convergence on datasets like ImageNet, and second because we believe the observations in the deterministic setting would be less significant due to being empirically unrealistic. A major surprise in our work is that RSI and EB remain remarkably well behaved despite stochastic gradients having high variance. Such observations would be much less significant in the full batch setting.

---

> > ### Comment · Reviewer_LuUd · 2023-11-22
> > **Response acknowledged**
> >
> > I thank the authors for their detailed response. I am satisfied with the answers provided, and I would like to keep my positive score.

---

### Official Review · Reviewer_eK7x · 2023-11-05

**Soundness:** 3 good
**Presentation:** 3 good
**Contribution:** 2 fair
**Rating:** 3
**Confidence:** 4

**Summary:**

Given the efficacy of stochastic first-order optimization algorithms on the non-convex landscapes of neural networks, this paper aims to show the simplicity of neural landscapes by tracking two key quantities that occur in the convergence analysis. Namely, they consider the restricted secant inequality (RSI) and the error bound (EB) during the optimization trajectory, whose ratio in turn boils down to the cosine similarity between the sampled minibatch gradient and the difference between the current and the final weights. Their observations seem to suggest that this quantity remains almost positive throughout (and that's why the titular phrase "no wrong turns") and has consistent behaviour during training. The results are demonstrated on residual networks in vision and language, and the effects of batch size, longer training duration, and initialization are considered.

**Strengths:**

Firstly, I think it's a great research direction to understand how the important quantities that occur in the optimization analyses actually behave in the neural landscapes and then ground up uncover properties of the loss landscape. These identified properties, such as the cosine similarity, might reflect an important aspect of the trajectories taken by 1st-order optimization algorithms in loss and the geometry of the landscapes in general. It is interesting to see how these quantities behave across various established empirical settings.

**Weaknesses:**

- **The nature of cosine similarity in high dimensions:** One of the fundamental issues about this paper is to what extent cosine similarity is a meaningful measure in high dimensions, which here are in the orders of tens of millions. The particular value of cosine similarity is somewhat difficult to grasp as a result, without the lack of a baseline that contextualizes its value. While the curves look alright and pretty stable throughout, the moment one looks at the value at the y-axis (which, e.g., is in the order of 1e-3 for ImageNet), doubts emerge if one is reading too much signal in the noise. This is in addition to the fact that the cosine similarity is between vectors that at their core are based on the same network structure will tend to have a higher cosine similarity (contrast that with arbitrary random vectors in the same dimensions).

   All of this makes it hard to be convinced by their argument about the stability of this cosine similarity (and optimization proceeding at a regular pace), when this value of cosine similarity almost hints at near orthogonality.

&nbsp;

- **The lack of clear insights that can be drawn:** Consider Figure 2, middle column, where the results are shown for ResNet50/ImageNet. It is unclear if anything much is being said by these graphs, except towards the end where it hints at being close to convergence. And that too, because the denominator in these expressions $\|w-w^\ast\|$ starts becoming small. More generally, a lot of these plots contain a sense of vagueness about the actual insights that can be drawn from them. The paper says there are "predictable" properties, but a less glorified way of phrasing would be they are somewhat banal.

&nbsp;

- **Dependence of the results on the chosen empirical setup** It is not clear if the results are overly reliant on the particular empirical setup chosen in the paper. Does the same thing hold when training with large learning rates? Bigger or smaller weight decay coefficients? When different learning rate schedulers are used? Essentially, my guess would be that something that would make the network jump in the optimization landscape (which is not uncommon in practical settings) should at least result in a different phenomenology than that presented here. Likewise, what happens when there is more noise in the dataset, say different amounts of label noise? Besides most of the experiments have been carried out with residual architectures. What happens if you consider a fully convolutional architecture, such as VGG on cifar10 and/or MobileNet on ImageNet?


---- Post rebuttal ----

I think the way of using cosine similarity currently is quite problematic and can thus be potentially misleading. The standard deviation for the cosine similarity of two independent random vectors on the sphere is itself, for dimension (and which seems, more or less, in the ballpark for the networks of presumably million parameters). But more than that, the vectors considered here are not purely arbitrary, and stem from the same network structure. For instance, Bao et. al (2023) consider as a baseline the cosine similarity of the network parameters at two different initializations, and then the calculations are contextualized based on such a value. Right now, I am afraid this paper is probably reading too much 'signal' in the noise, again due to a lack of a relevant baseline. I would recommend the authors contextualize with a measure like this and revise the results accordingly. At the moment, I will stick to my score.

Bao et al. 2023: Hessian Inertia in Neural Networks, ICML 2023 Workshop on High-dimensional Learning Dynamics

**Questions:**

See the weaknesses section.

Some minor questions:
- Influence of Batch size: Shouldn't the equation there be normalized since when you have a bigger batch size, you don't just add the gradients of smaller batch sizes but also normalize corresponding to their sizes? How does the argument then work out?
- Why should the set of global minima be convex? (page 3)
- Model width/depth: How are the relative trends when you normalize the quantities by the number of parameters?
- loLR: I understand the aim of Figure 3, but can you port these LR schedules for a fresh training of the corresponding networks? What do you observe?

---

> ### Author Response · Authors · 2023-11-15
> **Rebuttal - Weaknesses**
>
> We thank the referee for their careful consideration and feedback.
>
> **Nature of the cosine similarity in high dimensions (Weakness 1):**
> Interpreting the significance of our observations is the most challenging component of our submission and the referee is right to point out that it is difficult to grasp.
>
> Regarding the low value of the cosine similarity (~1e-3 for ImageNet), we discuss it explicitly in page 6 (“Low Value of Cosine Similarity” paragraph). If the cosine similarity was stable at a high value, then SGD would converge at an unreasonably fast pace. It is thus expected that cosine similarity would be low on average. We encourage the reviewer to compare our observations to a very badly dimensioned quadratic, for which optimization is slow and difficult, but the dynamics are simple, stable, and efficiently captured by the curvature.
>
> The relationship between sampled gradients and $w-w^\star$ is also discussed in detail in all of page 8, where we compare to anisotropic random walks in very high dimension as the referee suggests.
>
> Finally, the referee seems to oppose our claim about the stability of the cosine similarity to its low value (near orthogonality). The stability of the cosine similarity is relevant to the regularity of the convergence (in the parameter space), while its value is relevant to the rate of that convergence. Thus we argue these observations are not contradictory. Optimization trajectories on neural nets effectively progress at a slow pace, but with surprising regularity, encountering no significant setbacks… or wrong turns.
>
> **Lack of clear insights that can be drawn (Weakness 2):**
> Considering the well known complexity of neural loss landscapes, observing such “banal” (as the referee puts it) patterns for fundamental quantities is, paradoxically, surprising. We refer to Figure 5 for contrasting examples of what the cosine similarities could look like for trajectories computed on functions that are either non-convex or stochastic. The significance of our observations is discussed in the “significance” paragraph of page 6. Moreover, the stepwise optimal learning rate is directly tied to the local value of RSI and EB. This banal behavior of RSI and EB (which we denote as “predictable”) is thus what allows banal learning rate schedules to work so well. Unpredictable variations, as exhibited in Figure 5, would make it impossible to design a near-optimal learning rate schedule.
>
> **Dependence of the results on the chosen empirical setup (Weakness 3):**
> While we haven’t explored every possible setting variation, we believe we are presenting a sufficiently varied set of experimental settings to make a convincing case that our observations are not a byproduct of a specific empirical setup. It is noteworthy that our experiments require us to complete two full training passes (one to compute $w^\star$ and one to compute RSI and EB, knowing $w^\star$), effectively doubling computation costs.
>
> In order to make our experiments as relevant as possible, we tried to reproduce as closely as possible empirical practice, including in the tuning of hyperparameters. That is why we only measured RSI and EB on the best learning rates obtained by grid search. We do see jumps both in the training and validation losses, as is typically the case in practice. However, it appears RSI and EB are remarkably resilient to such jumps.
>
> While all our ImageNet experiments indeed use residual architectures (which were convenient to isolate width and depth as contrasting factors), we also presented our experiments with a MLP, UNet, SegNet, and Transformer, with similar conclusions for all of them. Note that we also used different LR schedules across tasks (constant LR for language, linear warmup cosine decay for images, piecewise constant for segmentation), in line with typical practice. While our experiments certainly don’t cover all of relevant deep learning settings, and experimenting e.g. with label noise would be very interesting, we argue in the light of everything above and all the ablations in our submission that our work found consistent patterns across a sufficiently wide array of settings to be considered unspecific to our setup.

---

> > ### Author Response · Authors · 2023-11-15
> > **Rebuttal - Questions**
> >
> > **Q1:**
> > Both RSI and EB are homogeneous of degree 1 in the stochastic gradient. Thus normalizing the gradients (which would be natural to do as the referee pointed out) would lead to the same normalization on the right hand side, and cancel out when considering the cosine similarity, which is their ratio. (Another way to look at it is that the cosine similarity is invariant when multiplying the gradient by a constant factor, as can be understood from its geometric interpretation).
> >
> > **Q2:**
> > This assumption comes from the definition of RSI and EB in prior works, which requires a convex set of minima to define the orthogonal projection used in their expression. For the local version of RSI and EB used throughout our work, which are taken wrt a fixed $w^\star$, we do not require any assumption on the set of minima.
> >
> > **Q3:**
> > We discuss this aspect in the “Model Depth and Width” paragraph at the top of page 7. Our findings (which we experimented only with ImageNet) is that RSI and EB barely change even when we multiply the width by 2, nearly quadrupling parameters, while they both decrease with the depth (see Figure 9 in appendix). EB decreases faster than RSI since their ratio, the cosine similarity, increases with depth (except for the first couple epochs where it is the opposite, see Figure 1 top left). Since both RSI and EB see negligible changes when we add width, we argue there is no need to normalize the number of parameters by e.g. adjusting the width.
> >
> > **Q4:**
> > We have not tried using the exact loLR schedule for a fresh training run. Our reasoning is that by changing the LR schedule, we change the speed of convergence, and thus we change ‘where we are’ (on the x-axis) on the loLR curve. In other words, the scale of the x-axis depends on the current LR schedule, but the general shape is preserved, and can be used to guide the functional form of the schedule (but not its exact hyperparameters). For instance, our first experiments on ImageNet used a piecewise constant schedule, and the loLR curve clearly suggested a linear warmup into cosine decay functional form, which became standard in this setting after several years of experimentation. In our opinion, being able to retrieve the functional form of standard learning rate schedules support the relevance of RSI/EB to empirical learning dynamics.

---

### Official Review · Reviewer_oKQj · 2023-11-09

**Soundness:** 3 good
**Presentation:** 3 good
**Contribution:** 3 good
**Rating:** 8
**Confidence:** 4

**Summary:**

This paper studies why stochastic first-order optimization methods can succeed in modern deep neural networks, despite the non-convex nature of the training problem. Specifically, this paper considers the ratio between (modified) Restricted Secant Inequality (RSI) and (modified) Error Bound (EB). This ratio turns out to be exactly the cosine similarity between G(w) and w-w*, where G(w) is the stochastic version of the local gradient, and w-w* is the direction from the converged point to the current point. In short, the considered ratio measures how much the local update direction aligns with the true direction pointing to the global optimum.

The authors conduct experiments in different settings to examine how this ratio performs and evolves during the network training. Surprisingly, this ratio always stays positive and is very stable during the training. This means that the landscape of modern neural networks, despite its non-convexity, is benign in the sense that every local update is positively correlated with the desired direction.

**Strengths:**

1. This paper proposed an effective metric, based on a variant of quantities involved in RSI and EB, to evaluate the training trajectory of modern neural networks. Both RSI  and EB are theoretically grounded metrics in optimization theory, capturing important properties of the optimization landscape.

2. Experiments validate that the cosine similarity metric stays positive and stable during training. This is an interesting finding, which may potentially inspire theoretical landscape study and/or convergence analysis of neural networks. Further, the trends of cosine similarity can partially justify the empirically used learning rate schedule.

3. This paper conducts extensive experiments under different settings and carries out comprehensive discussion. Notably, the proposed metric is dependent on the optimization trajectory, and thus the authors carefully discuss potential factors that might affect the interpretation of empirical results.

**Weaknesses:**

1. The paper does not provide any further theoretical analysis based on their empirical findings. Although existing works have demonstrated that well-behaved RSI and EB can lead to the convergence of first-order algorithms, it is not clear whether trajectory-dependent RSI and EB properties can lead to similar results.

2. The paper does not provide practical guidance on network training. Most conclusions are explanatory.

Overall, it is good to point out the "no wrong returns" property of the neural network landscape. However, its immediate impact on theory or practice is unclear to me.

**Questions:**

1. In contrasting examples, what is w*? The (known) global optimum or the converged point?

2. The authors claimed "these observed properties are sufficiently expressive to theoretically guarantee linear convergence" in the abstract. However, the paper considers variants of RSI and EB. Only local but not global properties are examined. Thus, the empirical findings cannot guarantee linear convergence. Is it correct?

---

> ### Author Response · Authors · 2023-11-15
> **Rebuttal**
>
> We thank the referee for their careful consideration and feedback.
>
> **Theoretical guarantees for local versions of RSI and EB (Weakness 1 and Question 2):**
> Interestingly, the linear convergence guarantees induced by RSI and EB only hinges on the local values of RSI and EB in the iterates that are explored. While previous theoretical works indeed only considered RSI and EB as global properties, Eq. (4) only depends on the local values of RSI and EB. The optimal learning rate $\eta^\star$ in Eq. (5) [which we will rename $\eta^\star(w)$ to clarify that it is dependent on $w$], and the corresponding step-wise guarantee in Eq. (6), only depend on the local value of RSI and EB. To obtain the guarantee of Eq. (7), it is thus sufficient for RSI and EB to be bounded on explored iterates.
>
> Alternatively, the interpolation conditions derived in “Gradient Descent is Optimal Under Lower Restricted Secant Inequality and Upper Error Bound” (NeurIPS 2022) state that as long as (local) RSI and (local) EB (wrt to some $w^\star$) are bounded by $\mu$ and $L$ on a finite set of points and gradients (e.g. the iterates), then there exists a function $f$ for which RSI and EB are bounded globally by $\mu$ and $L$, that exactly interpolates these points and gradients. Our trajectory is thus also a valid trajectory on $f$, and convergence guarantees obtained from global properties naturally convert to local properties for iterates.
>
> Considering the referee’s remark, we agree that the above points are not sufficiently clear. We propose to add a short paragraph at the end of Section 3 to emphasize that bounds on (local) RSI and (local) EB for iterates provide the same guarantees as global bounds. (Same for bounds on cosine similarity with adequate lr schedules).
>
> **$w^\star$ in contrasting examples (Question 1):**
>
> We use the converged point to be in the same setting as our experiments. We will specify this explicitly.
>
> **Lack of practical guidance (Weakness 2):**
>
> We absolutely agree with the referee. We believe that optimization of neural networks has been largely guided by intuitive engineering instead of theory in recent times, mostly because the classes of functions involved are insufficiently understood for relevant theory to emerge. We believe our work to be a promising – but incomplete – step towards understanding the structure of neural loss landscapes most relevant to 1st order optimization. Our work is indeed explanatory, but we hope that after further contributions to our understanding of empirical neural loss properties, we will eventually be able to develop a more adequate theory leading to useful algorithmic prescriptions !
> Perhaps the most immediate impact on theory is that we believe our work supports that RSI and EB are much more relevant assumptions to use in optimization theory for neural networks than e.g. smoothness and strong convexity.

---

> > ### Comment · Reviewer_oKQj · 2023-11-23
> >
> > Thanks for the response. I would like to keep my score.

---

### Meta-Review · Area_Chair_oXqb · 2023-12-11

**Metareview:**

The paper studies the neural network training process, by considering the ratio of two quantities, the Lower Restricted Secant Inequality (RSI) as well as an Upper Error Bound (EB). Their ratio turns out to be the cosine similarity of the current stochastic gradient and the direction from the current point to the optimum (converged point). An interesting observation stated is that cosine similarity always stays positive and stable during training, hence the title, even under noise from smaller batches.

While reviewers liked the topic and several ideas based on the two key quantities, concerns remained unfortunately also after the response by authors. Some concerns include the question of more principled foundations of the stated claims, and in general on using cosine similarity in this highly noise setting.

We hope the detailed feedback helps to strengthen the paper for a future occasion.

**Justification For Why Not Higher Score:**

The author response cleared up some things but unfortunately didn't convey enough to weigh up the main concerns raised by the two more negative reviewers at this point

**Justification For Why Not Lower Score:**

N/A

---

### Decision · Program_Chairs · 2024-01-16

Reject